# Identifying molecular features that are associated with biological function of intrinsically disordered protein regions

**Taraneh Zarin[1][†]\*, Bob Strome[1], Gang Peng[1], Iva Pritišanac[1,2], Julie D Forman-Kay[2,3], Alan M Moses[1]\***

[1]Department of Cell and Systems Biology, University of Toronto, Toronto, Canada; [2]Program in Molecular Medicine, Hospital for Sick Children, Toronto, Canada; [3]Department of Biochemistry, University of Toronto, Toronto, Canada

**Abstract** In previous work, we showed that intrinsically disordered regions (IDRs) of proteins contain sequence-distributed molecular features that are conserved over evolution, despite little sequence similarity that can be detected in alignments (Zarin et al., 2019). Here, we aim to use these molecular features to predict specific biological functions for individual IDRs and identify the molecular features within them that are associated with these functions. We find that the predictable functions are diverse. Examining the associated molecular features, we note some that are consistent with previous reports and identify others that were previously unknown. We experimentally confirm that elevated isoelectric point and hydrophobicity, features that are positively associated with mitochondrial localization, are necessary for mitochondrial targeting function. Remarkably, increasing isoelectric point in a synthetic IDR restores weak mitochondrial targeting. We believe feature analysis represents a new systematic approach to understand how biological functions of IDRs are specified by their protein sequences.

**\*For correspondence:**
taraneh.zarin@mail.utoronto.ca (TZ);
alan.moses@utoronto.ca (AMM)

**Present address:** [†]Systems Biology Program, Centre for Genomic Regulation (CRG), Barcelona, Spain

## Introduction

Intrinsically disordered regions (IDRs) are increasingly appreciated as playing key roles in diverse aspects of cell biology (*Forman-Kay and Mittag, 2013*). Bioinformatics methods identify thousands of IDRs in eukaryotic proteomes (*Dosztányi et al., 2005*; *Uversky, 2002*), and methods have been developed to predict biophysical or structural behavior for specific subsets of IDRs, such as those that fold upon binding (*Katuwawala et al., 2019*), contain specific N- or C-terminal motifs (e.g., *Chen et al., 2008*; *Chuang et al., 2012*), or phase separate (e.g., *Vernon et al., 2018*, reviewed in *Vernon and Forman-Kay, 2019*). Although there have been recent advances in predicting function from disordered region sequences (reviewed in *Uversky, 2020*), for the vast majority of IDRs, it is currently difficult to obtain high-specificity predictions of biological function or to identify which amino acid residues within them are important for function (*van der Lee et al., 2014*). Progress toward these goals would facilitate designing assays and mutagenesis experiments to understand the function of IDRs, which is especially pressing for those IDRs that are mutated in disease (*Vacic and Iakoucheva, 2012*).

For folded regions, similarity to homologous protein sequences offers highly specific and diverse predictions of biological function (*El-Gebali et al., 2019*) and can point to key residues (*Ondrechen et al., 2001*). However, IDRs often show little sequence similarity detectable in alignments, and therefore, alignment-based methods (*Davey et al., 2012*; *Nguyen Ba et al., 2012*) reveal only a small part of the functional elements in these regions (*Nguyen Ba et al., 2012*). Recent studies indicate that despite little conservation in amino acid sequence alignments, IDRs contain sequence-distributed molecular features, such as biophysical properties, repeats, and short linear

motifs, that are likely under natural selection and associated with biological function (*Zarin et al., 2019*; *Zarin et al., 2017*). This suggests that, using appropriate statistical methods, it should be possible to predict function for individual IDRs based on their amino acid sequences and to associate specific molecular features with biological functions. If successful, such an approach would allow general predictions of diverse functions for IDRs and generate testable hypotheses about molecular features that are important for those functions. This contrasts with current bioinformatics approaches that are designed to predict specific IDR subtypes or approaches that predict whole-protein function based on amino acid sequences (*Lobley et al., 2007*; *Radivojac et al., 2013*).

Here we describe a first general approach to predict diverse functions for IDRs and to identify molecular features that are important for those functions. Using this approach, we identify molecular features that are associated with functions such as subcellular organization and signaling. This allows us to confirm experimentally that molecular features are necessary for mitochondrial targeting signals and can be used to establish a mitochondrial targeting signal in a synthetic IDR. We also demonstrate how our approach can predict a role in transcription for the N-terminal IDR in an uncharacterized yeast protein Mfg1, based in part on evolutionary variation in polyglutamine repeats. Based on these examples, our work indicates a path forward for prediction of specific biological functions based on IDR sequences, and experimental characterization of molecular features that are necessary for IDR function.

## Results

### A model that predicts protein function from molecular features in IDRs

One of the main challenges of training statistical models to predict IDR function is the lack of systematic data at the IDR level. Functional annotations such as Gene Ontology (*The Gene Ontology Consortium, 2019*) or deletion phenotypes are at the protein level, but many proteins contain more than one predicted IDR. We therefore devised a statistical model that could combine the contributions of all IDRs in a protein (*Figure 1*, *Figure 1—figure supplement 1*). We assume that IDRs are associated with molecular feature vectors (e.g., the 'evolutionary signatures' that summarize the evolution of molecular features [*Zarin et al., 2019*]), and that functional annotations are binary at the protein level (we treat each annotation independently): either the protein is associated with the given annotation or is not. If we knew which IDR in the protein was responsible for the function, this would amount to a standard classification problem.

Since, in fact, we do not know which IDR is responsible for the functional annotation, we treat this as a hidden variable. To ensure that the model infers only a few important molecular features for each function, we use LASSO regularization (*Hastie et al., 2015*) and to infer the hidden variables, we use an expectation–maximization (E–M) algorithm (see Materials and methods). The model automatically determines which IDRs are likely to be responsible for each function and which features are most informative for each function (*Figure 1*). We refer to this model as FAIDR (Feature Analysis of Intrinsically Disordered Regions). Given IDR feature vectors, in principle, it can predict protein function, infer which IDR is responsible for a given function, and determine the molecular features that are important for each function.

### IDR function can be predicted from protein-level annotations and IDR sequence properties

We trained FAIDR models using protein-level functional annotations from the Saccharomyces Genome Database (SGD) (*Cherry et al., 2012*) (see Materials and methods). As previously described (*Zarin et al., 2019*), to summarize the molecular features of IDRs, we used evolutionary signatures, vectors of Z-scores that summarize the deviations of 82 molecular features in disordered regions from the distribution of those features obtained from simulations of IDR evolution. For each molecular feature, we calculate the mean and (log) variance across a set of orthologous IDRs, which leads to a total of 164 features describing the evolution of these molecular features in each IDR. Here we wanted to use both the evolutionarily conserved and diverged parts of the disordered region to form the evolutionary signature. Therefore, in contrast to our previous work (*Zarin et al., 2019*), we

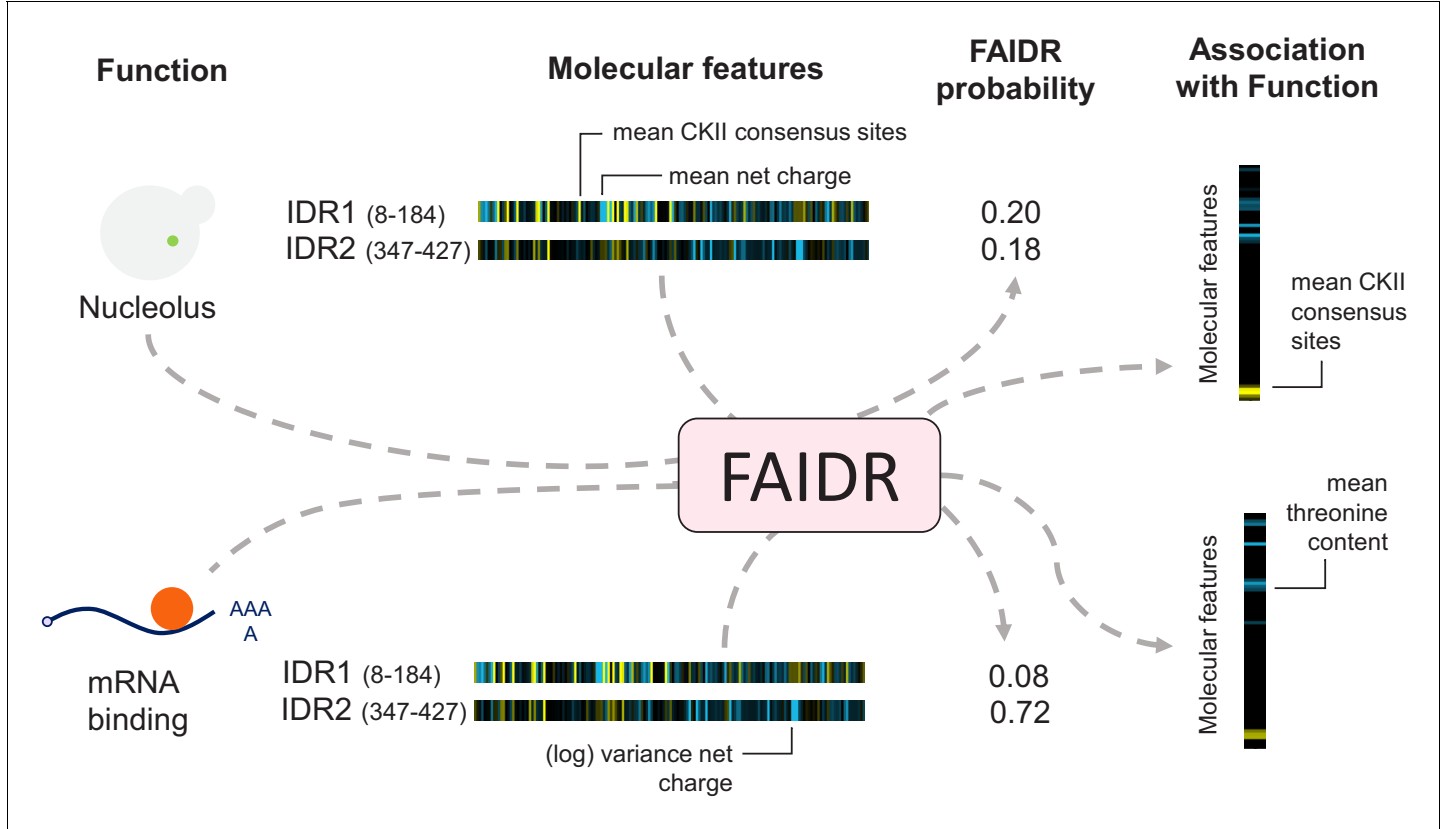

**Figure 1.** Schematic of the FAIDR statistical model. Example inputs (function, molecular features) and outputs (FAIDR probability, association with function) are shown. Given a set of features (e.g., molecular features comprising evolutionary signatures, as shown) for one or more IDRs in a given protein and functional annotation information (e.g., whether or not the protein is annotated with mRNA binding function or nucleolus function, as shown), FAIDR outputs the probability that the IDR is associated with the given function as well as the strength and direction of the association with function for each molecular feature. Molecular features comprising the evolutionary signatures are represented by a heatmap scaling from blue (decreased mean or variance of the feature compared to our null expectation of IDR evolution) to yellow (increased mean or variance of the feature compared to our null expectation of IDR evolution). The association with function is represented by a heatmap scaling from blue (negative association) to yellow (positive association). For example, for nucleolus function (as shown), mean CKII consensus sites are strongly positively associated with this function (indicated with yellow), whereas for mRNA binding (also shown), mean threonine content is negatively associated with this function (indicated with blue).

The online version of this article includes the following figure supplement(s) for figure 1:

**Figure supplement 1.** The FAIDR probabilistic model.

did not constrain the simulations to preserve short conserved motifs (see Materials and methods for details). The evolutionary signatures that we calculated for the yeast proteome are available for download and visualization at http://www.moseslab.csb.utoronto.ca/idr/.

To demonstrate the power of FAIDR at predicting IDR-level function, we used two systematic, global screens for function associated with IDRs that are mapped to protein coordinates: mitochondrial N-terminal targeting signals identified in a proteome-wide screen (*Vögtle et al., 2009*) and high confidence Cdc28 phosphorylation sites (which point to kinase substrates that are key regulatory sites) curated from in vitro and in vivo studies (*Lai et al., 2012*). Importantly, these datasets are unseen by the model during training and testing, and because they are mapped to protein coordinates, they can be used to create IDR-level functional annotations (see Materials and methods). To further ensure that there is no 'leakage' between training and test data, we fit the model parameters (*b*, described above) to protein-level annotations from 80% of the proteome and evaluate the performance on the unseen IDR-level data for the 20% of proteins that were not part of the training dataset. We find that the held-out predictions from FAIDR do as well or better than state-of-the-art predictors of mitochondrial targeting signals and Cdc28 substrates that were specifically designed

to predict these functions (Mitofates and Condens, respectively) (*Fukasawa et al., 2015*; *Lai et al., 2012*; *Figure 2*). This suggests that our general approach is capturing comparable functional information to previous function-specific predictors.

For functions and phenotypes with no IDR-level data (the vast majority), we tested whether we could identify molecular features predictive of IDR function by evaluating the area under the curve (AUC) in cross-validation on held-out proteins (see Materials and methods). We found a diverse set of protein annotations and phenotypes that could be predicted with reasonable power (overall average fivefold cross-validation AUC = 0.81, *Figure 2—figure supplement 1*, gray bars), ranging from well-characterized IDR functions such as signal transduction and mitochondrial targeting, to underappreciated IDR functions such as ribosome biogenesis and DNA damage (full list of functions in *Supplementary file 1*). Although some of these annotations and phenotypes are correlated, this analysis strongly supports the idea that there is rich functional information in IDR sequences (*Zarin et al., 2019*), and is evidence that the model is not overfitting.

Because evolutionary signatures can be technically difficult and time consuming to compute (*Zarin et al., 2019*), we also tested whether FAIDR could predict function from single-species profiles of molecular features. Indeed, we found that the 82 sequence features (*Zarin et al., 2019*) computed from *Saccharomyces cerevisiae* IDR sequences alone (scaled to unit variance and 0 mean) could also predict protein function (overall average fivefold cross-validation AUC = 0.77, *Figure 2—figure supplement 1*, unfilled bars). Although the overall power was significantly less than the evolutionary signatures (paired t-test over 23 classes p<0.001), for some IDR functions the predictive power was similar, which suggests that IDR function can be predicted from single-protein sequences.

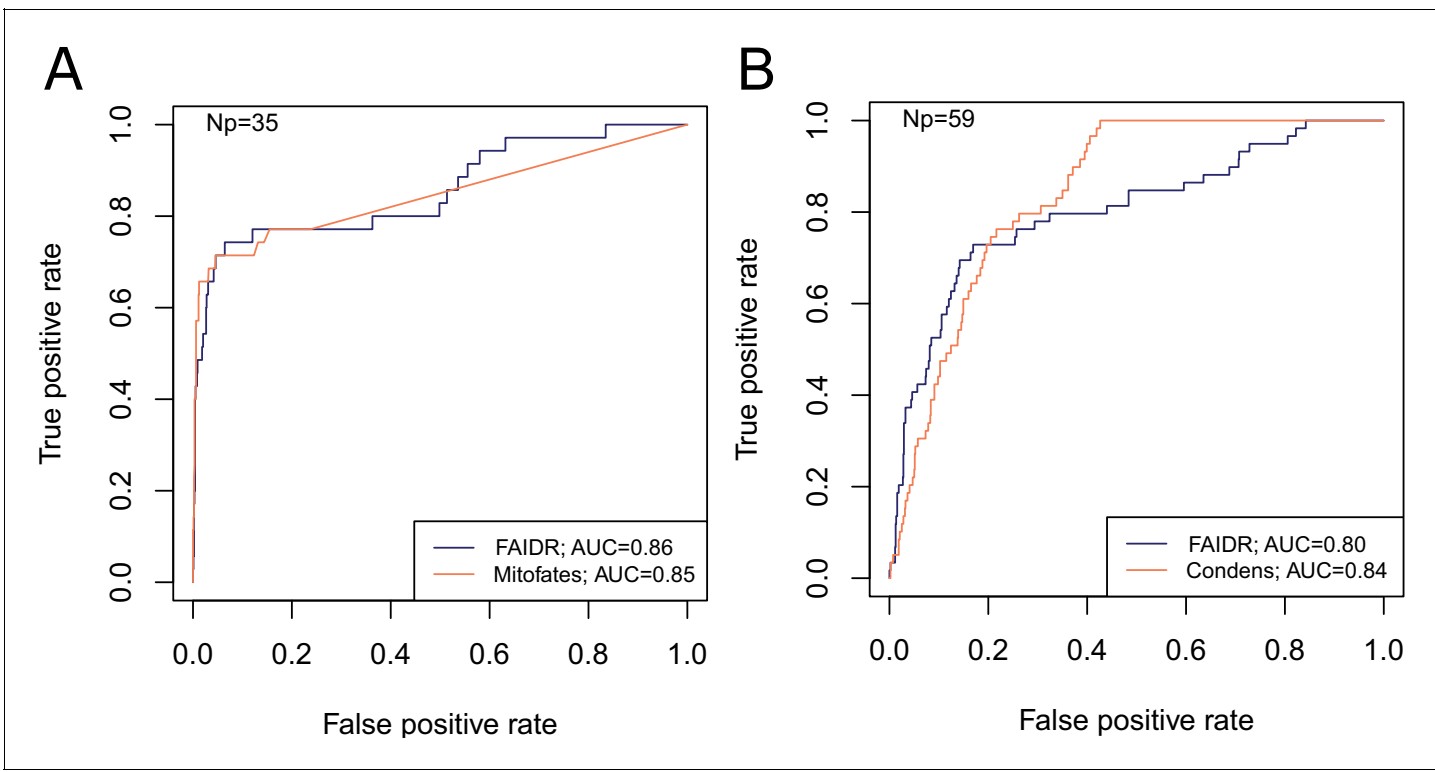

**Figure 2.** FAIDR trained on evolutionary signatures of IDRs can predict unseen data comparably to state-of-the-art, specific predictors for mitochondrial targeting signals (Mitofates [*Fukasawa et al., 2015*]) and Cdc28 substrates (Condens [*Lai et al., 2012*]). Receiver operating curves (ROC) on a held-out 20% sample are shown for FAIDR trained on evolutionary signatures (blue) versus Mitofates (**A**) and Condens (**B**) (orange). Area under curve (AUC) is indicated in legend on bottom right for each method. The number of positive IDRs (Np) in the held-out 20% is indicated in the top left of each plot. The online version of this article includes the following source data and figure supplement(s) for figure 2:

**Source data 1.** Data for ROC curves including IDR coordinates, FAIDR probabilities for held out IDRs and ground truth labels from experimental data mapped to each IDR.

**Figure supplement 1.** Predicting diverse protein function and phenotype using FAIDR.

Identifying informative molecular feature descriptions for IDRs that are easier to compute than evolutionary signatures is a promising area for further research (see Discussion).

## Exploring molecular features of IDRs associated with diverse functions and phenotypes

We next sought to determine which molecular features were associated with IDR function. To do so, we obtained the features that were assigned non-zero coefficients using FAIDR, and fit a standard logistic regression to that subset of features. We extracted the associated t-statistics and visualized them as a clustered heatmap (*Figure 3A*). Encouragingly, we found that related functions group together in this analysis (*Figure 3A*, indicated by symbols above function names). For example, consistent with their putative functions (*Darling et al., 2018*), proteinaceous membraneless

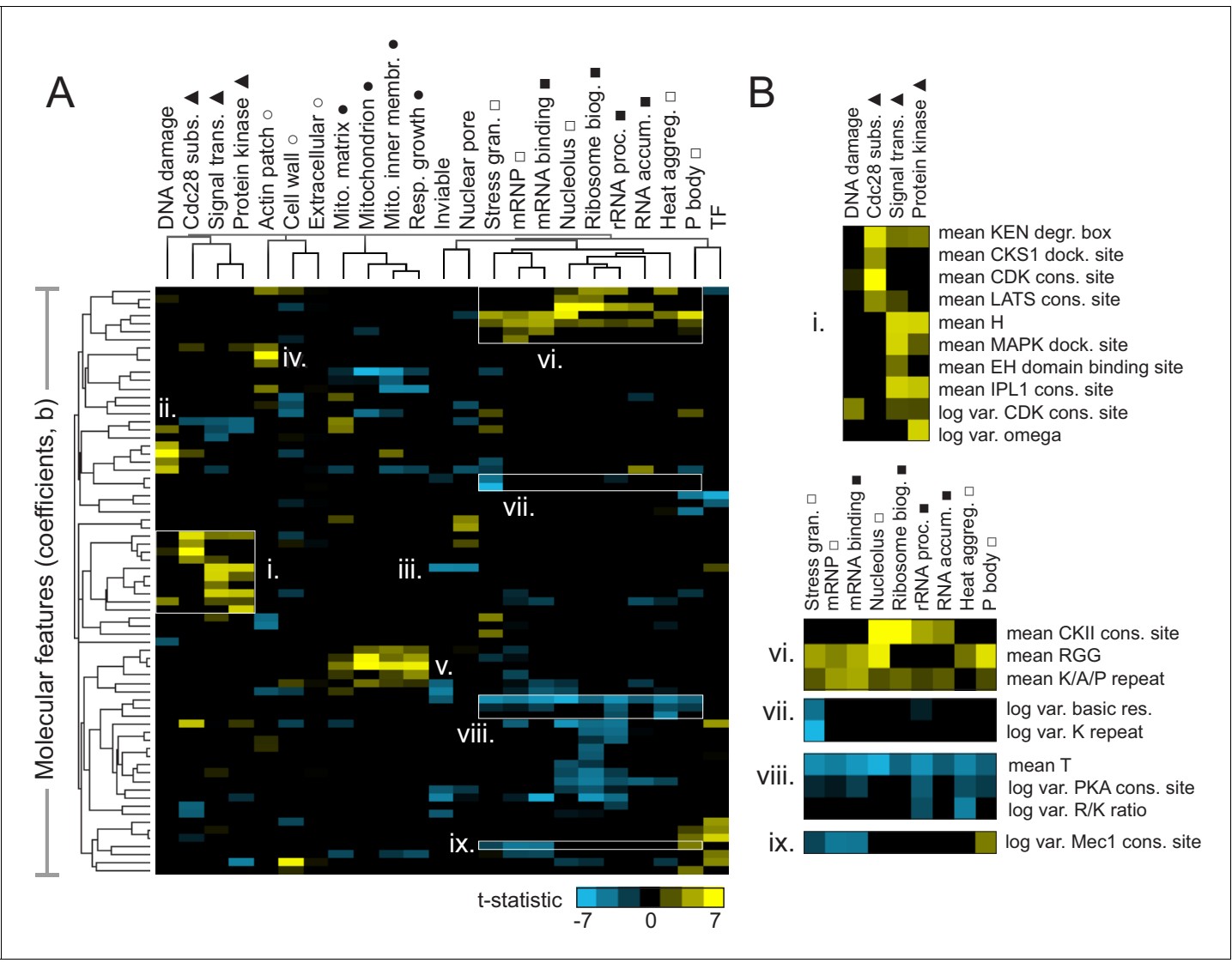

**Figure 3.** Molecular features of IDRs are associated with specific functions. (**A**) Hierarchical clustering of t-statistics obtained from regression of 23 functions and phenotypes on evolutionary signatures (means and variances of 82 molecular features). Symbols beside function/phenotype names indicate related functions/phenotypes. Indicated subgroups (i–ix) are referred to and described in the main text. (**B**) Examples of positive and negatively predictive molecular features for signaling (i) and PMLOs (vi-ix).

The online version of this article includes the following source data for figure 3:

**Source data 1.** Raw T-statistics used for cluster analysis and display.

organelles (PMLOs) (*Figure 3A*, unfilled square) clustered with several phenotypes and functions related to RNA (*Figure 3A*, filled square).

We note that the features that underlie these t-statistics represent the extent to which the mean or (log) variance of a sequence-distributed molecular feature (such as a physicochemical property or amino acid repeats) differs from our null expectation over evolutionary time (see Materials and Methods for details). It is relatively straightforward to interpret a positively predictive mean of a property such as hydrophobicity to indicate that elevated hydrophobicity is associated with a certain function. It is more difficult to interpret how the variance of a sequence-distributed property over evolution can be interpreted with respect to its function. For both mean and variance of sequence-distributed properties, we interpret deviations from our null expectation to point to the properties that are (positively or negatively) associated with function (see Discussion).

We found that several molecular features known to be associated with specific biological functions were recovered in this analysis. For example, signaling-associated functions such as Cdc28 kinase substrates, signal transduction, and protein kinase activity share an increase in KEN degradation boxes (*Figure 3B,i*), while Cdc28 substrates (as expected) are distinguished by increased CKS1 docking sites and CDK consensus sites (*Figure 3B,i*). The DNA damage response seems to share some features with the more general functions discussed above, but is also (as expected) associated with the presence of specific motifs: PCNA/Pip boxes, Mec1/Tel1 consensus sites (*Lai et al., 2012*), and NLSs (nuclear localization signals) (*Figure 3A,ii*). Signal transduction and protein kinase activity share increased histidine residues (*Figure 3B,i*), which we speculate is related to histidine kinase signaling. Interestingly, the opposite (a decrease in histidine residues) is predictive of essentiality ('inviable' phenotype) and nuclear pore localization (*Figure 3A,iii*). Other features that are consistent with known disordered region function include the presence of PRK1 consensus sites as a predictor of actin cortical patch function (*Figure 3A*,iv) (*Sekiya-Kawasaki et al., 2003*), and increased isoelectric point as a predictor of mitochondrial functions (*Figure 3A,v*) (reviewed in *Jaussi, 1995*).

We next focused on features predictive of PMLOs, as it is currently unclear whether the association of IDRs with these organelles is determined by amino acid sequence. As expected, we found the presence of RGG (*Chong et al., 2018*) and K/A/P repeats (*van der Lee et al., 2014*) among the strongest predictors for all these compartments (*Figure 3B,vi*). We also found a decrease in threonine residues associated with these, as well as all the RNA-related functions (*Figure 3B,viii*), which to our knowledge has not been previously reported. We were particularly interested in features that could discriminate between PMLOs. We found that stress granules are distinct from all other bodies, showing a decrease in the variance of basic residues and lysine repeats (*Figure 3B,vii*). Nucleoli and associated ribosomal functions are distinct in that they show increased CKII consensus sites (*Figure 3B,vi*). We also found that processing body (P body) proteins show an increased variance in Mec1 consensus sites, while mRNPs and mRNA binding proteins show a decrease in this feature (*Figure 3B*,ix).

## Molecular features in the Cox15 IDR predictably affect mitochondrial targeting of Cox15

In order to directly test whether the features in our evolutionary signatures that are predictive of function are required for function (and not simply correlated with function), we sought to manipulate these features in an example IDR. For this, we chose the N-terminal IDR of the Cox15 protein, which is a known mitochondrial targeting signal (*Vögtle et al., 2009*). As in previous work (*Zarin et al., 2019*), there are several reasons that make this system ideal for testing our hypotheses about molecular features that underlie a biological function: (1) localization to mitochondria is a clearly observable phenotype and (2) mitochondrial precursor peptides contain well-characterized sequence features that enable their prediction as mitochondrial targeting sequences (*Fukasawa et al., 2015*) (see Discussion). Using FAIDR, we first identified the top predictive features for this IDR (Cox15 IDR 1, which spans amino acids 1–45 in Cox15) that indicate association with mitochondrial localization ('mitochondrion' GO annotation), and plotted the molecular feature vector (evolutionary signature) values corresponding to these top molecular features in *Figure 4A*. The top predictive molecular feature (a positive predictor) is mean isoelectric point (pI), which also has a relatively high Z-score in the Cox15 IDR 1 feature vector. This indicates that increased isoelectric point is not just a generally positive predictor for mitochondrial localization, but that elevated isoelectric point is present in the

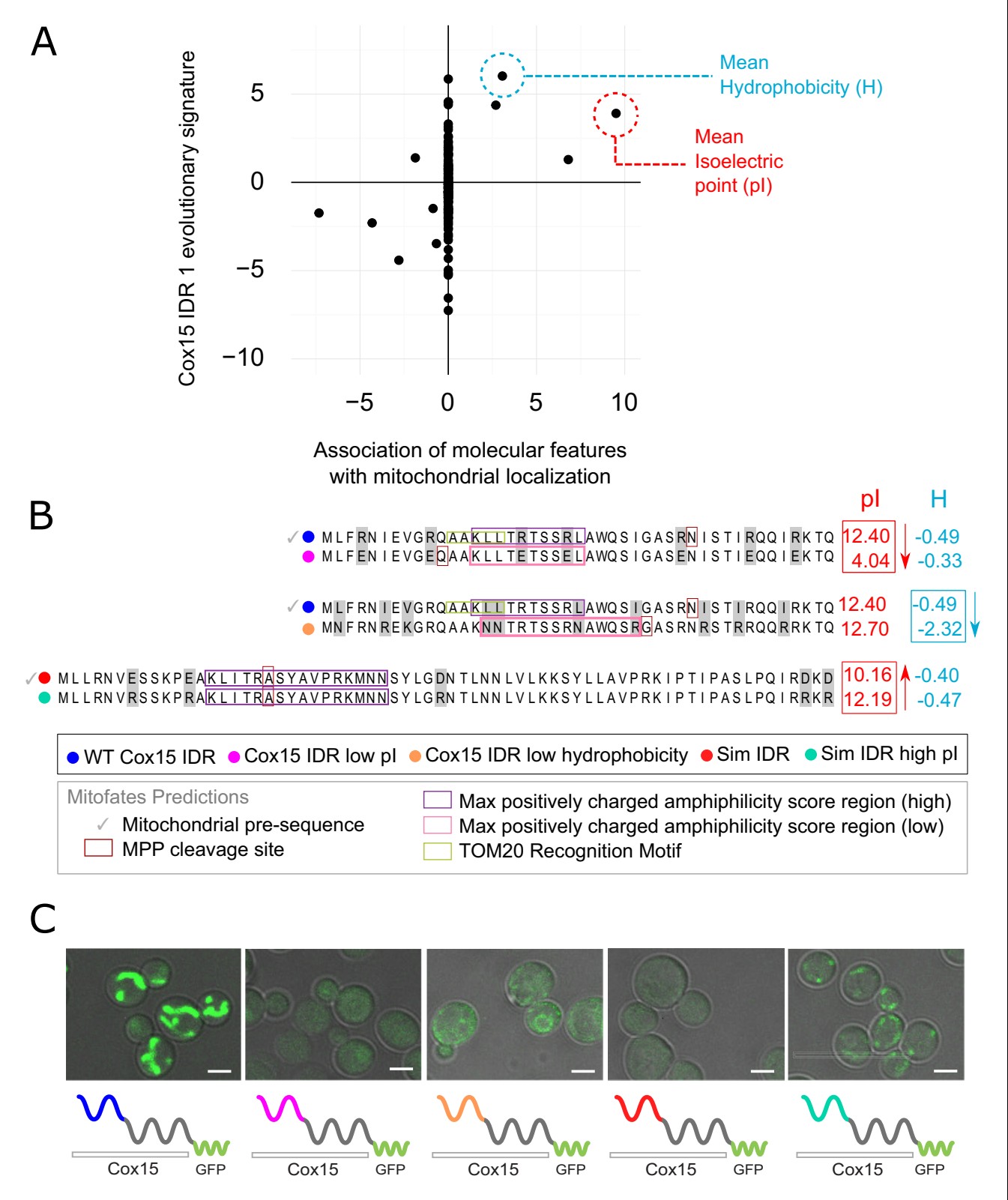

**Figure 4.** Molecular features predicted to be associated with 'mitochondrion' GO annotation affect mitochondrial localization phenotype. (A) Feature vector of Z-scores (evolutionary signature) of the Cox15 N-terminal IDR (Cox15 IDR 1) (y-axis) is plotted against predictive features (t-statistics) for mitochondrial function as determined by FAIDR (x-axis). Two of the top features associated with mitochondrial localization are mean isoelectric point (pI), circled in red, and mean hydrophobicity (H), circled in blue. (B) Amino acid sequences for each Cox15 IDR variant. Wild-type and simulated Cox15

*Figure 4 continued on next page*

*Figure 4 continued*

IDR sequences are compared to the same sequence with mutations altering isoelectric point (pI) and hydrophobicity. Sequences with variable residues (gray) are visualized with Jalview (*Waterhouse et al., 2009*). Predictions from Mitofates (*Fukasawa et al., 2015*), a predictor of mitochondrial targeting sequences, are also shown: prediction of whether or not the given sequence is a mitochondrial pre-sequence (check mark), as well as the predicted location of the cleavage site in the wild-type and mutated sequences (red box), the regions predicted to have a high (purple box) or low (pink box) max positively charged amphiphilicity score, and region of the TOM20 (receptor) recognition motif (green box). (C) Micrographs showing the mitochondrial localization phenotype for different budding yeast strains that differ in their Cox15 N-terminal IDRs. Green shows GFP-tagged Cox15 localization. Left to right: wild-type IDR, Cox15 IDR with low pI, Cox15 IDR with low hydrophobicity, simulated IDR, simulated IDR with high pI. Scale bar represents 1 µm.

The online version of this article includes the following figure supplement(s) for figure 4:

**Figure supplement 1.** Micrographs showing (columns, left to right) bright-field, DAPI, Cox15-GFP, and 4× zoomed Cox15-GFP/DAPI overlay of different budding yeast strains with different Cox15 IDR genotypes (labels on left side of image).

---

Cox15 IDR specifically. Another top positive predictor that is present in this IDR is increased hydrophobicity (*Figure 4A*).

In order to manipulate these sequence-distributed features (isoelectric point and hydrophobicity), we constructed strains with different Cox15 IDR 1 genotypes and molecular features (*Figure 4B*). We first created a mutant IDR with lower pI by mutating each arginine residue (R) in the wild-type IDR sequence to a glutamic acid (E), which lowered the pI of the wild-type IDR from 12.40 to 4.04 while only minimally changing the hydrophobicity (*Figure 4B*, top). We tested mitochondrial localization in all GFP-tagged Cox15 strains using co-localization with mitochondrial DNA as visualized by DAPI staining (as in *Higuchi-Sanabria et al., 2016*; *Figure 4—figure supplement 1*). As expected, the wild-type GFP-tagged Cox15 strain has a clear mitochondrial localization (*Figure 4C*, left; *Figure 4—figure supplement 1*). However, the 'Cox15 IDR low pI' strain no longer localizes to the mitochondria (*Figure 4C*, *Figure 4—figure supplement 1*). This provides evidence that the elevated pI is necessary for mitochondrial targeting function. Similarly, we created a mutant IDR with lower hydrophobicity by mutating each leucine residue (L) to asparagine (N), each valine (V) to lysine (K), and each isoleucine residue (I) to arginine (R) (*Figure 4B*, middle). This changed the three most hydrophobic residues into the three least hydrophobic according to the Kyte–Doolittle scale (*Kyte and Doolittle, 1982*), while only minimally affecting the isoelectric point. Interestingly, the 'Cox15 IDR low hydrophobicity' strain also fails to localize to the mitochondria (*Figure 4C*, *Figure 4—figure supplement 1*), which indicates that elevated hydrophobicity is also necessary for mitochondrial targeting function.

To test whether elevated pI is sufficient to restore a targeting signal, we first constructed an N-terminal Cox15 IDR in which we 'knocked out' sequence-distributed features that we predict to be important for specific IDR function. To make this synthetic Cox15 IDR, we simulated the evolution of the Cox15 N-terminal IDR under a model that randomly evolves IDRs while preserving position-specific variation in evolutionary rates, but includes no specific constraints on the sequence-distributed molecular features used to train FAIDR (as in *Iserman et al., 2020*; *Figure 4B*, bottom). We note that this simulated IDR is not a completely random or generic IDR; for example, the simulated IDR shares 4/5 of the N-terminal amino acids with the wild-type Cox15 IDR from which it 'evolved' in silico (*Figure 4B*). Nevertheless, consistent with the functional importance of the sequence-distributed features, the strain in which the N-terminal IDR of GFP-tagged Cox15 is replaced with this simulated IDR fails to show a mitochondrial localization pattern (*Figure 4C*, *Figure 4—figure supplement 1*), indicating that the first few conserved amino acids in the IDR, the predicted cleavage site and helicity (*Figure 4B*) are not sufficient for mitochondrial localization. We next made mutations in the simulated Cox15 IDR to increase the pI back to the range of the wild-type Cox15 N-terminal targeting signal. We mutated each glutamic acid (E) and aspartic acid (D) residue to arginine (R), bringing the pI from 10.2 in the simulated IDR to 12.2 in the simulated IDR with high pI. Remarkably, this mitochondrial localization phenotype seems to be partially rescued when the pI of the simulated IDR is increased ('sim IDR high pI', *Figure 4C*, *Figure 4—figure supplement 1*). This indicates that the elevated pI in the context of a synthetic, non-functional Cox15 N-terminal disordered region is sufficient to restore weak mitochondrial targeting function.

Taken together, these results demonstrate that functional associations with evolutionary signatures revealed by the FAIDR statistical model can be used to design mutations in IDRs that alter

phenotype in predictable ways, and suggest that a small number of molecular features may be necessary and sufficient for partially restoring some IDR functions.

## Inferring function for specific IDRs in uncharacterized proteins

One of the important use cases of our approach (the FAIDR model trained on evolutionary signatures of predicted IDRs) is to predict functions for specific protein regions that are otherwise challenging, if not impossible, to annotate using current methods. To demonstrate the use of FAIDR for annotating protein regions, we chose to focus on open reading frames (ORFs) annotated as 'uncharacterized' in the yeast proteome by SGD (*Cherry et al., 2012*). There are currently 739 uncharacterized ORFs in the yeast proteome, of which 121 have one or more IDRs for which we are able to make predictions. We extracted the probability that each IDR corresponds to each of the functions that we considered (*equation 1*), and used this to make functional predictions for IDRs in uncharacterized ORFs.

One interesting example is the N-terminal IDR in putative mitochondrial protein SHH3, which we predict to be associated with the mitochondrion, as well as the mitochondrial inner membrane. Indeed, SHH3 assumes what looks to be a mitochondrial localization in recent localization collections such as the LoQAtE (Localization and Quantitation Atlas of the yeast proteomE) database (*Breker et al., 2014*; *Breker et al., 2013*) and Yeast RGB (*Dubreuil et al., 2019*). Another interesting case is an IDR in MFG1, which FAIDR predicts to be associated with sequence-specific DNA binding (*Figure 5A*), indicating that MFG1 could be a transcription factor. Indeed, all other IDRs with a higher probability for association with transcription factor activity according to FAIDR are IDRs from confirmed transcription factors (Sok2, Sfp1, Mot3) (*Figure 5A*). Despite being an uncharacterized ORF, Mfg1 regulates filamentous growth by interacting with the FLO11 promoter and regulating FLO11 expression (*Ryan et al., 2012*), and has a nuclear localization (*Huh et al., 2003*), further supporting the idea that it is indeed a transcription factor. Using FAIDR, we can specifically identify the Mfg1 protein region that is associated with transcription factor activity (*Figure 5B*). Furthermore, by looking at the top predictive features that FAIDR points to (*Figure 5C*, left) and plotting the feature vector (evolutionary signature) for these top features, we can begin to understand the molecular features that are contributing to this function for the Mfg1 IDR 1 specifically. For example, increased variance of glutamine residues in Mfg1 is a positive predictor of its association with transcription factor activity (*Figure 5C*, orange box), and this sequence feature can be seen in a multiple sequence alignment of the Mfg1 IDR and its orthologs (*Figure 5C*, right). Thus, we can formulate the hypothesis that this specific sequence feature is important for function, and test this by modulating the number of glutamine repeats in the lab (see Discussion).

## Discussion

We have demonstrated a systematic approach to associate molecular features of IDR sequences with diverse biological functions. Our method stands in contrast with those established for specific subtypes of IDRs, as well as existing methods that use amino acid sequences to predict whole-protein function. Our goal is to generate hypotheses about both the function of individual IDRs and which molecular features in those IDRs are important for those functions, much like Pfam annotations provide both predictions of molecular functions of domains and hypotheses about specific residues within those domains that are important for that function (*El-Gebali et al., 2019*).

In the case of Mfg1, which is a protein of unknown function, the N-terminal IDR is among the IDRs most strongly associated with transcription factor function in the yeast proteome. Because Mfg1 contains three predicted IDRs (and in general proteins contain multiple IDRs), this illustrates the advantage of our approach that infers which (if any) of the IDRs are associated with a particular function. Although in some well-studied cases, it is clear that the C-terminal and N-terminal IDRs have different functions (e.g., p53, *Krois et al., 2018*; *Laptenko et al., 2016*), in general we do not know if functional information within IDRs is modular (confined to one IDR) or distributed (spread over all the IDRs in a protein). In our exploration of yeast IDRs strongly associated with functions, we identified interesting cases where multiple IDRs within a protein were strongly associated with different functions (e.g., *Figure 5—figure supplement 1*) possibly supporting the idea that IDRs can be thought of as functional modules. However, we note that the identification of IDR boundaries in protein sequences (via bioinformatics or experiments) is an error-prone process (*Nielsen and Mulder,*

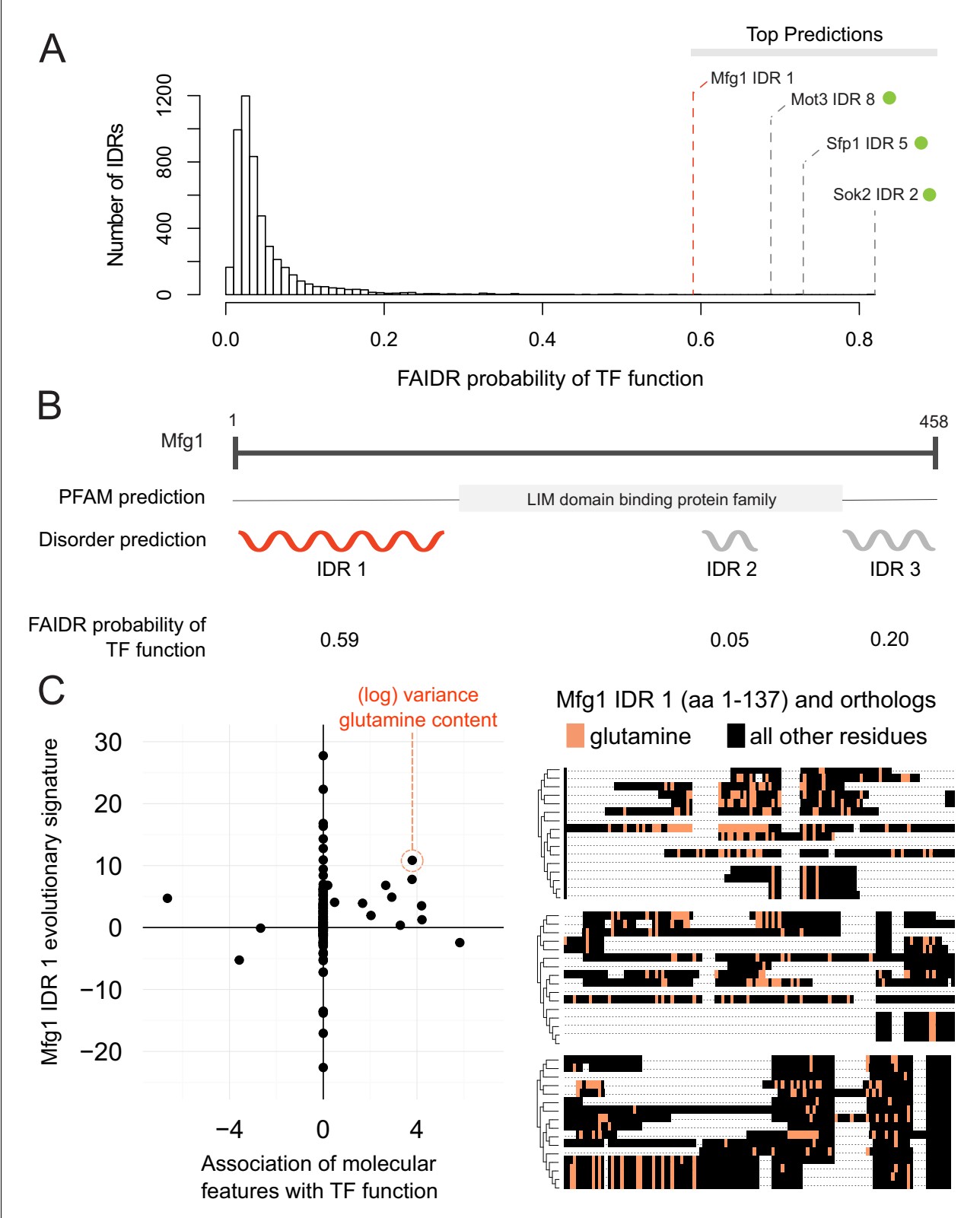

**Figure 5.** Identification of a specific IDR sequence associated with transcription factor activity in an uncharacterized protein. (**A**) Histogram of probabilities for association of IDRs with sequence-specific DNA binding (transcription factor or 'TF') function. Top predictions are indicated. From the top predictions, IDRs from known transcription factors are indicated with a green dot. Mfg1 is an uncharacterized protein. (**B**) Mfg1 protein coordinates, with all known domain annotations (via PFAM), disorder prediction (via DISOPRED3), and FAIDR probability for association with sequence-

*Figure 5 continued on next page*

*Figure 5 continued*

specific DNA binding indicated. (**C**) Feature vector (evolutionary signature) for Mfg1 IDR 1 (y-axis) is plotted against predictive features for transcription factor/sequence-specific DNA binding function as determined by FAIDR (x-axis). Log variance in glutamine (Q) residues is highlighted as one of the top features (orange circle) associated with TF function. Glutamine residues (Q) are indicated (in orange) in the IDR of Mfg1 and orthologs obtained from the YGOB (**Byrne and Wolfe, 2005**). All other (non-glutamine) residues are indicated in black. Multiple sequence alignment of orthologs is visualized using Jalview (**Waterhouse et al., 2009**). Species in alignment (from bottom to top) are as follows: *Saccharomyces cerevisiae*, *Saccharomyces mikatae*, *Saccharomyces kudriavzevii*, *Saccharomyces uvarum*, *Candida glabrata*, *Kazachstania africana*, *Kazachstania naganishii*, *Naumovozyma castellii*, *Naumovozyma dairenensis*, *Zygosaccharomyces rouxii*, *Torulaspora delbrueckii*, *Eremothecium (Ashbya) gossypii*, *Eremothecium (Ashbya) cymbalariae*, *Lachancea kluyveri*, *Lachancea thermotolerans*, and *Lachancea waltii*.

The online version of this article includes the following figure supplement(s) for figure 5:

**Figure supplement 1.** Example of modular IDR function predicted by FAIDR.

*2019*), and thus in many cases, we are not confident how many IDRs a protein 'truly' contains. A key technical challenge hindering inference of the IDR sequence features that are associated with function is that most proteome-scale functional information is at the level of entire proteins and proteins often contain multiple predicted IDRs, the exact boundaries of which are often uncertain. Further improvement of the approaches described here could allow simultaneous inference of IDR function and boundaries, perhaps providing more insight.

The analysis of Mfg1 also reveals that homologs of the N-terminal IDR show elevated variance in glutamine repeats, a property of IDRs that we find associated with transcription factor function, consistent with previous reports (**Freiman and Tjian, 2002**; **Gemayel et al., 2015**; **Klemsz and Maki, 1996**). Thus, in addition to providing a hypothesis about which IDR is important for Mfg1 function, we predict that modulation of the glutamine residues in the N-terminal IDR of Mfg1 will impact its function in transcription. We note that in the case of Mfg1, the molecular feature is an elevation of evolutionary variance, and this feature cannot be modulated experimentally because we only affect one single value. Instead, we interpret the associations with our evolutionary signatures as pointing toward testable hypotheses. While FAIDR analysis using descriptions of molecular features based on single sequences would be easier to interpret, we found that our features did not perform as well when computed on *S. cerevisiae* alone (**Figure 2—figure supplement 1**). This supports the use of evolutionary features in this case, but as we discuss further below, this suggests that development of improved molecular features that can be manipulated in single sequences is an area for future research.

In addition to other previously reported molecular features (the expected short linear motifs and repeats associated with signaling, posttranslational regulation, and RNA binding), we found an unexpected depletion of threonine residues in IDRs in most classes of RNA-binding proteins and an unexpected decreased variance of lysine residues specific to stress-granule proteins. Future experiments will test the importance of these features. Nevertheless, our interpretable molecular features and statistical framework both confirm previous associations with function and reveal new properties of IDRs, while still providing predictive power comparable to current function-specific bioinformatics approaches.

Understanding which molecular features in IDRs are necessary and sufficient for various biological functions is an important area for further research. At least in the case of the Cox15 N-terminal mitochondrial targeting signal, we could show that two molecular features, mean isoelectric point and mean hydrophobicity, the increase of which we found to be positively associated with function, appear to be necessary for function of that IDR (**Figure 4**). We note that hydrophobicity and isoelectric point are only two of the top features that we chose to focus on in this study and are not the only two features that are important for mitochondrial localization. It is known that features including positive charge, potential to form an amphipathic helix, hydrophobicity, and the presence of cleavage motifs can help identify mitochondrial targeting signals (**Fukasawa et al., 2015**; **Garg and Gould, 2016**; **Roise and Schatz, 1988**; **Vögtle et al., 2009**). We note that the molecular features that we find to be associated with mitochondrial targeting function are in line with previous findings and serve to illustrate how our method could be used to identify and test functional molecular features for protein regions that are less characterized than mitochondrial targeting signals. Furthermore, while mitochondrial pre-sequences are known to form helices upon interacting with their target receptor, they do not engage in a stable interaction and sample different conformations

(*Abe et al., 2000*; *Kohda, 2018*; *Saitoh et al., 2007*). We therefore include these regions in our analysis of IDRs. We suggest that our approach can be used to identify molecular features associated with function for any protein regions that are not amenable to standard methods of sequence analysis. To facilitate wider use of this approach, we have developed a website that provides search, visualization, and comparison of evolutionary signatures for predicted IDRs in the yeast proteome (http://www.moseslab.csb.utoronto.ca/idr/).

Remarkably, we also found that artificially increasing the isoelectric point of a simulated Cox15 N-terminal IDR (i.e., an IDR with no enrichment of sequence-distributed features) enabled us to create a weak synthetic targeting signal. To our knowledge, this represents the first synthetic IDR designed to perform a specific biological function and highlights the benefit of using a simple, interpretable statistical model: we could hand-design mutations in a synthetic IDR to perform a desired function through eight amino acid changes. In general, designing synthetic IDRs represents a new approach for testing the importance of molecular features.

In all, we found that diverse functions can be associated with the IDR sequence features included in our evolutionary signatures (*Zarin et al., 2019*). However, we note that these features of disordered regions were obtained from a review of the literature, and new features are continuously being discovered (e.g., patterning of aromatic residues [*Martin et al., 2020*]). It is likely that including these features will further improve predictive power of the approach described here. Nevertheless, these features are necessarily biased by current research interests, and because they rely on comparisons to simulations of molecular evolution, they are cumbersome to compute. In future work, we believe that application of advanced computational methods will enable more systematic exploration, ideally automatically learning molecular features that are most important for IDR function.

## Materials and methods

### The FAIDR statistical model

FAIDR was implemented in R with the glmnet package (*Friedman et al., 2010*). Code and results are available at https://github.com/taraneh-z/FAIDR; *Zarin, 2021*; copy archived at swh:1:rev:b1ef30705f7133f21201166522077a062984566b. A schematic of the model is provided in Supplementary materials (*Figure 1—figure supplement 1*).

We assume that IDRs are associated with molecular feature vectors Z (e.g., the 'evolutionary signatures' that summarize the evolution of molecular features [*Zarin et al., 2019*]) and that functional annotations, Y, are binary indicator variables at the protein level (we treat each annotation independently) such that Y = 1 if the protein is associated with the given annotation and Y = 0 if the protein is not associated with the given annotation. If we knew that the jth IDR in the ith protein was responsible for the function, this would amount to a standard classification problem.

Since our goal is to develop a method that can identify the molecular features associated with each function as well as to predict the function of the IDRs, we consider logistic regression, which summarizes the contribution of each feature to the prediction task using a vector of coefficients, b, one for each of m features. Using standard logistic regression, if the IDR responsible for a function were known, we could write the probability of a function given the features

$$P(Y_i = 1 | Z_{ij}, b) = \frac{1}{1 + e^{-Z_{ij}b}} = 1 - P(Y_i = 0 | Z_{ij}, b) \tag{1}$$

Since, in fact, we do not know which IDR is responsible for the functional annotation Y, we treat this as a hidden indicator variable, X, and marginalize. The likelihood of the data for n proteins is then:

$$P(Y | Z, b) = \prod_{i=1}^{n} \sum_{j=1}^{r_i} P(X_{ij} = 1) P(Y_i = 1 | Z_{ij}, b)^{Y_i} P(Y_i = 0 | Z_{ij}, b)^{1-Y_i} \tag{2}$$

where $r_i$ is the number of disordered regions in the ith protein and $X_{ij} = 1$ if the jth IDR in the ith protein is responsible for the function, Y.

To maximize the likelihood of our model (*Equation 2*), we use the E–M algorithm or iteratively maximize the expected complete log likelihood (reviewed in *Moses, 2017*). To derive the E–M algorithm for this model, we first write the complete likelihood (*CL*, the log likelihood assuming the hidden variables are observed):

$$CL = \prod_{i=1}^{n} \prod_{j=1}^{r_i} \left( \left[ \frac{1}{1+e^{-Z_{ij}b}} \right]^{Y_i} \left[ 1 - \frac{1}{1+e^{-Z_{ij}b}} \right]^{1-Y_i} \right)^{X_{ij}} = \prod_{i=1}^{n} \prod_{j=1}^{r} \left( [h(Z_{ij})]^{Y_i} [1-h(Z_{ij})]^{1-Y_i} \right)^{X_{ij}} \quad \text{(A.1)}$$

where $h(Z_{ij}) = \frac{1}{1+e^{-Z_{ij}b}}$.

Taking logs and expectations yields:

$$\langle log\, CL \rangle = \sum_{i=1}^{n} \sum_{j=1}^{r} \langle X_{ij} \rangle \left[ Y_i log\, h(Z_{ij}) + (1-Y_i) log(1-h(Z_{ij})) \right] \quad \text{(A.2)}$$

where the angle brackets denote expectations. In the E-step of the E–M algorithm, these expectations are calculated using Bayes' theorem as follows:

$$\langle X_{ij} \rangle = P(X_{ij} = 1 | Y_i, Z_{ij}, b) = \frac{P(X_{ij}=1)P(Y_i|X_{ij}=1, Z_{ij}, b)}{P(Y_i|Z_{ij}, b)} = \frac{Y_i h(Z_{ij}) + (1-Y_i)(1-h(Z_{ij}))}{\sum_k Y_i h(Z_{ik}) + (1-Y_i)(1-h(Z_{ik}))} \quad \text{(A.3)}$$

Note that in (*Equation A.3*), we have assumed no prior knowledge about which IDR in a protein is most likely to be responsible for the function (uninformative priors), $P(X_{ij}=1)$ is constant over *j*.

In the M-step, we maximize the expected complete log likelihood. Inspection of *Equation A.2* reveals exactly one term for each IDR, weighted by $\langle X_{ij} \rangle$, the expected value of the hidden variable. Thus, our model corresponds to an iteratively reweighted logistic regression problem, where the weights are the current guess of the hidden variable, $\langle X_{ij} \rangle$, representing which IDR is responsible for the function. We can therefore maximize the L1 penalized version of this expected complete log likelihood by adding an extra factor to the weights of the IRLS algorithm (*Hastie et al., 2015*). The modified weights are:

$$w_{ij} = \langle X_{ij} \rangle h(Z_{ij}) (1-h(Z_{ij})) \quad \text{(A.4)}$$

In each M-step in our implementation, we do five iterations of this IRLS (on standard LASSO regression) to obtain new parameter values. We used a fixed L1 penalty of 0.2 throughout, as we found this to provide the strong regularization needed for our small numbers of positive examples. To improve the numerical stability of IRLS (needed when we had small numbers of positive examples relative to the number of features), we also used a small L2 penalty (alpha = 0.99 in glmnet) and we monitored the weights (A.4) and set them to an arbitrarily small value if they reached 0. After five iterations of IRLS, we recalculate the expected values of the hidden variables $\langle X_{ij} \rangle$. We initialize $\langle X_{ij} \rangle$ by setting them equal to $1/r_i$, so that they are downweighted equally.

## Obtaining evolutionary signatures for IDRs in the yeast proteome

In order to calculate evolutionary signatures from predicted IDRs (disordered regions predicted from the *S. cerevisiae* proteome using DISOPRED3 [*Jones and Cozzetto, 2015*] that are 30 amino acids or longer), we used a similar method to previously published work (*Zarin et al., 2019*) with some modifications.

Briefly, for each set of orthologous IDRs (obtained from the Yeast Gene Order Browser [*Byrne and Wolfe, 2005*] and filtered to include those proteins for which there are 10 or more orthologs), we simulated 1000 sets of orthologous IDRs using previously described methods (*Nguyen Ba et al., 2014*; *Nguyen Ba et al., 2012*) and quantified the difference in the evolution of molecular features in the real set of orthologous IDRs to that of the simulated orthologous IDRs using a standard Z-score (as described in *Zarin et al., 2019*). The evolutionary signature of each IDR thus comprised a vector of 164 Z-scores mean and log variance over evolution for 82 molecular features (listed in *Zarin et al., 2019*). A negative Z-score implies that the evolution (in mean or variance) of a molecular feature is lower than our null expectation, while a positive Z-score implies that the evolution (in mean or variance) of a molecular feature is higher than our null expectation for the given

IDR. A Z-score around zero implies that the molecular feature in question is evolving according to our null expectation of IDR evolution. The null expectation includes selection to preserve amino acid residue frequency according to disordered region-specific substitution matrices, but no selection to preserve the sequence-distributed molecular features.

In previous work (*Zarin et al., 2019*), we calculated these evolutionary signatures for highly diverged segments of IDRs by constraining the evolution of short conserved segments in our simulations of IDR evolution. For this study, we wanted to include all of the potentially functional sequence information in our evolutionary signatures (i.e., highly diverged segments as well as short conserved segments) and therefore did not constrain the evolution of short conserved segments in the evolutionary simulations. The evolutionary simulation software (*Nguyen Ba et al., 2014*; *Nguyen Ba et al., 2012*) takes as input maximum-likelihood estimates for the column (site-specific) rate of evolution, the local rate of evolution, and whether or not a site is predicted to be part of a conserved short linear motif (*Nguyen Ba et al., 2012*). In order to ignore the constraint on these short conserved segments, we randomly scrambled the column (site-specific) rate of evolution for each IDR and set each found motif to 0. This results in evolutionary simulations of IDRs in which positionally conserved sites evolve with the same average constraint as other sites in the IDR.

Evolutionary signatures are available for download and visualization at http://www.moseslab.csb.utoronto.ca/idr/.

## Compiling functions and phenotypes for prediction

We compiled a series of functional annotations, phenotypes, and datasets to predict with our model. These fell into the following categories:

1. Gene Ontology annotations that we previously found to be strongly enriched in our unsupervised clustering analysis (*Zarin et al., 2019*), acquired from SGD (*Cherry et al., 2012*).
2. Datasets that screened the *S. cerevisiae* proteome for membraneless organelle/reversible protein assembly formation under stress (*Mitchell et al., 2013*; *Narayanaswamy et al., 2009*; *Wallace et al., 2015*).
3. A dataset of gold-standard Cdc28 substrate predictions (*Sharifpoor et al., 2011*).
4. A set of publicly available *S. cerevisiae* phenotype annotations from SGD (*Cherry et al., 2012*).

In total, we found 23 functions and phenotypes for which we could get accurate predictions (*Figure 2—figure supplement 1*). Overall, we tested the model on 120 functions and phenotypes.

## Evaluating FAIDR at the IDR level

We obtained two datasets for which we could define 'positive' and 'negative' IDRs from proteomics data.

First, we obtained a dataset of mitochondrial targeting signals from a global analysis that identified and validated mitochondrial targeting signals in highly purified mitochondria using combined fractional diagonal chromatography (*Vögtle et al., 2009*).

Second, we obtained Cdc28 phosphorylation sites discovered in high-throughput kinase-specific mass spectrometry experiments (*Holt et al., 2009*; *Lai et al., 2012*) and mapped them to IDRs. If at least one verified phosphorylation site was identified in a region, we considered this IDR to be a target of Cdc28.

To make predictions of functions for unseen IDRs, we used the parameters (*b*) estimated for each function and computed *Equation (1)* above. We compared these probability values to the known IDR function as the probability cutoff is varied. We used ROCR (*Sing et al., 2009*) to compute ROC curves and AUCs.

Even though these datasets were not used directly for the protein-level annotations, they may have been used as part of the evidence that led to the annotation of the protein. Therefore, we sought to further remove the potential for circularity by removing 20% of proteins from the training dataset and only evaluating the predictive power on the IDRs from those 20%. We report the number of positive IDRs in this held-out data set in *Figure 2*.

In order to confirm that the results of our IDR-level predictions are not based on sequence similarity between the training and test set, we performed an all-by-all protein BLAST (*Altschul et al., 1990*) with default parameters, using the training set as the 'query' and the test set as the 'subject'.

We found that out of 1016 IDRs in the test set, 49 (5%) had sequence similarity to IDRs in the training set (i.e., had a BLAST e-value that was less than $10^{-5}$). For mitochondrial targeting signals, this sequence similarity explains at most 3% of our predictive power (1/35 IDRs in the positive test set have sequence similarity to IDRs in the training set according to the above e-value threshold). For CDK targets, this sequence similarity explains at most 10% of our predictive power (6/59 IDRs in the positive test set have sequence similarity to IDRs in the training set according to the above e-value threshold). IDRs with sequence similarity tend to be in paralogous proteins.

## Evaluating FAIDR at the protein level

Although predicting protein function was not a primary goal of FAIDR, we evaluated the AUCs on held-out protein annotations to confirm that FAIDR could learn features that are predictive of a diverse set of IDR functions. To make predictions of functions for held-out proteins ($Y$), we sum the weighted contributions of all the IDRs:

$$P(Y_i = 1|Z_i, b) = \sum_{j=1}^{r} P(X_{ij} = 1|Z_{ij}, b) P(Y_i = 1|Z_{ij}, b) \qquad \text{(A.5)}$$

where $P(Y_i = 1|Z_{ij}, b)$ is given in **Equation (1)** above. To compute $P(X_{ij} = 1|Z_{ij}, b)$ for an unseen sequence, we cannot use (Equation A.3) because it treats $Y$ as an observed variable. Instead we used $P(X_{ij} = 1|Z_{ij}, b) = \frac{1}{r_i}$, which amounts to a simple average over the IDRs in each protein.

For each functional category/annotation, we used fivefold cross-validation, randomly dividing the data into 80% of proteins for training and 20% of proteins for testing. For each fold of cross-validation, we compute the AUC using ROCR (**Sing et al., 2009**) on the random 20% held-out data. We report the average AUC over the five held-out datasets.

## Clustering of t-statistics

We fit a standard logistic regression to molecular features that obtained non-zero coefficient values from FAIDR. We extracted the corresponding t-statistic for each molecular feature and clustered these t-statistics and the 23 functions that we predicted using hierarchical clustering for **Figure 3**. Prior to clustering, we assigned those molecular features with a coefficient of 0 a corresponding t-statistic of 0 and filtered the 164 molecular features to only include those molecular feature vectors that had at least one t-statistic value of 3 or greater for a predicted function. This filtering resulted in a total of 72 molecular features. We used the Cluster 3.0 program (**de Hoon et al., 2004**) to perform hierarchical clustering (clustering both 'genes' [molecular feature t-statistics] and 'arrays' [functions] using uncentered correlation distance, with calculated weights and using average linkage). We visualized the clustering using Java Treeview (**Saldanha, 2004**).

## Construction of Cox15-GFP strains with variable molecular features and confocal microscopy

We obtained a BY4741-derived Cox15-GFP-tagged budding yeast strain from the Yeast GFP collection (**Huh et al., 2003**), which serves as our wild-type IDR strain. We mutated the N-terminal IDR in this strain as previously described (**Zarin et al., 2019**) using the Delitto Perfetto method (**Storici et al., 2001**), which performs markerless site-directed mutagenesis in the genome. In order to construct the IDR mutant strains ('Cox15 low pI', 'Cox15 low hydrophobicity', 'Sim IDR', and 'Sim IDR high pI'), we used gene synthesis (via Integrated DNA Technologies [IDT]) whose sequences we confirmed using PCR and Sanger sequencing. The genotypes for the IDR strains are as follows:

WT:
ATGCTTTTCAGAAACATAGAAGTGGGCAGGCAGGCAGCTAAGCTATTAACGAGAACC
TCGAGTCGTTTGGCCTGGCAAAGTATTGGGGCCTCAAGGAATATTTCTACCATCAGACAA-
CAAATCAGAAAGACTCAA
Cox15 low pI:
ATGCTTTTCGAAAACATAGAAGTGGGCGAACAGGCAGCTAAGCTATTAACGGAAACC
TCGAGTGAATTGGCCTGGCAAAGTATTGGGGCCTCAGAAAATATTTCTACCATCGAACAA-
CAAATCGAAAAGACTCAA
Cox15 low hydrophobicity:

```
ATGaatTTCAGAAACagaGAAaagGGCAGGCAGGCAGCTAAGaataatACGAGAACCTCGAG
TCGTaatGCCTGGCAAAGTagaGGGGCCTCAAGGAA
TagaTCTACCagaAGACAACAAagaAGAAAGACTCAA
```
Sim IDR:
```
ATGCTGCTGAGAAACGTTGAATCCTCCAAACCCGAAGCAAAACTAATTACCAGAGCTTC
TTACGCCGTGCCCAGGAAAATGAACAATTCATACTTGGGCGATAATACATTGAATAACC
TGGTCTTAAAGAAGAGCTATCTTTTAGCTGTTCCCAGAAAGATTCCCACGATTCCAGCCAG
TCTGCCGCAAATTCGTGACAAGGAT
```
Sim IDR high pI:
```
ATGCTGCTGAGAAACGTTAGATCCTCCAAACCCAGAGCAAAACTAATTACCAGAGCTTC
TTACGCCGTGCCCAGGAAAATGAACAATTCATACTTGGGCAGAAATACATTGAATAACC
TGGTCTTAAAGAAGAGCTATCTTTTAGCTGTTCCCAGAAAGATTCCCACGATTCCAGCCAG
TCTGCCGCAAATTCGTAGAAAGAGA
```

Confocal microscopy (Leica TCS-SP8, 63× oil immersion objective, Leica HyD hybrid detectors) was used to image cells. Overnight cultures were diluted 1/10 in fresh SD-His media and grown to log phase for 3 hr at 30°C. Where applicable (*Figure 4—figure supplement 1*), cells were treated with 0.5 µg/ml DAPI in SD-complete growth media for 15 min. Cultures were concentrated 1/10 by centrifugation at 3000 rpm for 3 min and imaged on standard, uncoated glass slides. Nuclear and mitochondrial localization was imaged sequentially to eliminate potential crosstalk in the following series: DAPI (excitation/emission 358 nm/461 nm), GFP (excitation/emission 488 nm/507 nm), and a bright-field image. Each strain was imaged on at least three separate days. Images in *Figure 4C*, *Figure 4—figure supplement 1* were taken on the same day with the same microscope settings across all images.

# Acknowledgements

We thank Alex X Lu for comments on the manuscript and Dr Alan R Davidson for discussions about IDR functional predictions. We thank Dr Helena Friesen and Dr Brenda Andrews for providing strains from the yeast GFP collection. We thank Canadian Institutes for Health Research (CIHR) for funding to AMM and JDF-K (grant no. PJT-148532), the Canada Research Chairs program and a CIHR Foundation grant (grant no. FDN-148375) to JDF-K, Canada Foundation for Innovation (CFI) for funding to AMM, and the Natural Sciences and Engineering Research Council of Canada (NSERC) for funding to TZ.

# Additional information

### Competing interests

Alan M Moses: Reviewing editor, *eLife*. The other authors declare that no competing interests exist.

### Funding

| Funder | Grant reference number | Author |
| --- | --- | --- |
| Canadian Institutes of Health Research | PJT-148532 | Julie D Forman-Kay Alan M Moses |
| Canadian Institutes of Health Research | FDN-148375 | Julie D Forman-Kay |
| NSERC | | Alan M Moses |
| Fondation canadienne pour l'innovation | | Alan M Moses |
| NSERC | Canada Graduate Scholarship | Taraneh Zarin |
| Hospital for Sick Children | RESTRACOMP Research Fellowship | Iva Pritišanac |

The funders had no role in study design, data collection and interpretation, or the decision to submit the work for publication.

### Author contributions

Taraneh Zarin, Conceptualization, Software, Investigation, Visualization, Methodology, Validation, Writing - original draft, Writing - review and editing; Bob Strome, Investigation, Methodology; Gang Peng, Software, Visualization; Iva Pritišanac, Investigation, Writing - review and editing; Julie D Forman-Kay, Conceptualization, Funding acquisition, Writing - review and editing; Alan M Moses, Conceptualization, Resources, Software, Supervision, Funding acquisition, Validation, Methodology, Writing - original draft, Project administration, Writing - review and editing

### Author ORCIDs

Taraneh Zarin (ID) https://orcid.org/0000-0003-1253-3843
Julie D Forman-Kay (ID) https://orcid.org/0000-0001-8265-972X
Alan M Moses (ID) https://orcid.org/0000-0003-3118-3121

### Decision letter and Author response

Decision letter https://doi.org/10.7554/eLife.60220.sa1
Author response https://doi.org/10.7554/eLife.60220.sa2

# Additional files

### Supplementary files

• Supplementary file 1. Table of function and phenotype abbreviations used in this article, corresponding function and phenotype full names, references, and function and phenotype name used in data frames/source data (i.e., name with no spaces).

• Transparent reporting form

### Data availability

All data generated or analysed during this study are included in the manuscript, supporting files and the accompanying website.

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

## Appendix 1

We first review logistic regression and IRLS and then explain how our approach for associating protein functions with sequence features in disordered regions is a simple extension of logistic regression.

### Multiple LASSO logistic regression

For classification, the target Y (or dependent variable in regression) is one if true and 0 if false for each of $n$ observations. More formally,

$$Y = Y_1, \, Y_2, \ldots, Y_n$$

$$Y_i \in \{0, 1\}$$

We assume that for each observation $Y_i$, we have a vector of features that we would like to use for prediction, say, $X_i$.

$$X_i = X_{i1}, \, X_{i2}, \ldots, X_{im}$$

Since we are doing linear regression, we define a linear combination of weights, $b$, and features, $X$:

$$b_0 + b_1 X_{i1} + b_2 X_{i2} + \, \ldots \, + \, b_m X_{im} = b_0 + \sum_{j=1}^{m} X_{ij} b_j \equiv X_i b$$

For (binary) classification, we use the logistic function to turn the linear combination into a probability

$$P(Y_i = 1 | X_i, b) = \frac{1}{1 + e^{-X_i b}}$$

and we know that $P(Y_i = 0 | X_i, b) = 1 - P(Y_i = 1 | X_i, b)$ because there are only two possibilities for $Y_i$, and the probability must sum to one.

The likelihood (the probability of $Y$ given the model) is the product over the (by assumption) independent and identically distributed (i.i.d) observations, because the probability of a series of observations is just the probabilities of independent events multiplied together:

$$P(Y | X, b) = \prod_{i=1}^{n} \left[ \frac{1}{1 + e^{-X_i b}} \right]^{Y_i} \left[ 1 - \frac{1}{1 + e^{-X_i b}} \right]^{1 - Y_i}$$

In the expression above, we have used the trick that $x^0 = 1$ and $x^1 = x$, so when $Y_i = 1$, the first term, which is $P(Y_i = 1 | X_i, b)$, is included in the product and the second term is just 1 (because $1 - Y_i = 0$), so it does not change the product. Similar logic means that the second term, which corresponds to $P(Y_i = 0 | X_i, b)$, is selected when $Y_i = 0$. Maximum-likelihood estimation means choosing the values (or estimates) of $b$ that maximize the likelihood function.

In practice, we work with the log likelihood (whose maximum is also the maximum of the likelihood):

$$logL = \sum_{i=1}^{n} Y_i \log \frac{1}{1 + e^{-X_i b}} + (1 - Y_i) \log \left( 1 - \frac{1}{1 + e^{-X_i b}} \right)$$

LASSO logistic regression adds a penalty to the log likelihood, so that

$$PlogL = \sum_{i=1}^{n} Y_i \log \frac{1}{1 + e^{-X_i b}} + (1 - Y_i) \log \left( 1 - \frac{1}{1 + e^{-X_i b}} \right) - \lambda \sum_{j=1}^{m} |b_j|$$

This penalty ensures that in order to have a non-zero value, a parameter $b$ contributes at least $\lambda$ to the model fit (in units of log likelihood). $\lambda$ is a hyperparameter and cannot be estimated on the training data. In practice it can be chosen through cross-validation or based on prior knowledge.

Although there is no closed form for the values of $b$ that maximize the penalized loglikelihood (or standard loglikelihood), the problem can be solved by a so-called 'iteratively reweighted least squares' (IRLS) algorithm, which converts the logistic regression problem to a 'weighted' ordinary least squares (OLS) regression. In the LASSO or penalized likelihood framework, instead of OLS regression, L1 penalized regression is done in each iteration (*Tibshirani et al., 2015*).

## IRLS: weighted OLS regression for logistic regression

In general, if the weights are defined to be $w = w_1, w_2, \ldots, w_n$, then the 'weighted' OLS regression problem is to choose $b$ so that it minimizes

$$\sum_{i=1}^{n} w_i (Y_i - X_i b)^2$$

In the case of IRLS for logistic regression, it is known that for a certain choice of weights and transformation of $Y$, the OLS solution will be exactly the maximum-likelihood solution for logistic regression. The weights are given by

$$w_i = \frac{1}{1 + e^{-X_i b}} \left( 1 - \frac{1}{1 + e^{-X_i b}} \right)$$

and the transformed variable, say, $Y^*$, is defined as:

$$Y_i^* = X_i b + \frac{Y_i - \frac{1}{1 + e^{-X_i b}}}{w_i}$$

In these equations, $b$ is our current estimate of the parameters. Since the weights and transformed target variable depend on the current estimate of the parameters, the algorithm proceeds by alternately solving the weighted OLS to estimate the parameters, and then updating the weights and transformed target variable. This proceeds iteratively from an initial guess, but in practice is known to converge rapidly.

## A model to account for multiple disordered regions in one protein

If we let $Y_i$ represent the function of the proteins that we would like to predict, we would like to combine the features for all IDRs, $Z_i = (Z_{i1}, Z_{i2}, \ldots Z_{ir})$, for that protein. A simple assumption we can make is that one of the IDRs is responsible for the function, and the others are irrelevant. We use a hidden variable, $X_{ij}$, to represent this, so that if $X_{ij} = 1$, the $j$th IDR in the $i$th protein is the one responsible for the function. In a linear regression framework, the likelihood of this model is

$$P(Y|Z,b) = \prod_{i=1}^{n} \sum_{j=1}^{r} P(X_{ij}) \left[ \frac{1}{1 + e^{-Z_{ij}b}} \right]^{Y_i} \left[ 1 - \frac{1}{1 + e^{-Z_{ij}b}} \right]^{1-Y_i}$$

In practice, to maximize likelihoods involving hidden variables, we use the E–M algorithm. This means iteratively maximizing the 'expected complete log likelihood', where we assume the hidden variables are observed, and equal to their expectations given our current parameter estimates. To derive this, we first write the complete likelihood, the log likelihood assuming the hidden variables are observed:

$$CL = \prod_{i=1}^{n} \prod_{j=1}^{r} \left( \left[ \frac{1}{1 + e^{-Z_{ij}b}} \right]^{Y_i} \left[ 1 - \frac{1}{1 + e^{-Z_{ij}b}} \right]^{1-Y_i} \right)^{X_{ij}}$$

Once again, the exponent $X_{ij}$ ensures that terms corresponding to the IDRs that are not responsible for the function do not contribute to the likelihood. Taking logs and expectations yields

$$\langle \log CL \rangle = \sum_{i=1}^{n} \sum_{j=1}^{r} \langle X_{ij} \rangle \left( Y_i \log \frac{1}{1 + e^{-Z_{ij}b}} + (1 - Y_i) \log \left( 1 - \frac{1}{1 + e^{-Z_{ij}b}} \right) \right)$$

where the angle brackets denote expectations. In the M-step of the E–M algorithm, we maximize the expected complete log likelihood. Comparing the standard loglikelihood for logistic regression (where we replace X with Z for consistency):

$$logL = \sum_{i=1}^{n} Y_i \log \frac{1}{1 + e^{-Z_i b}} + (1 - Y_i) \log \left( 1 - \frac{1}{1 + e^{-Z_i b}} \right)$$

Inspection of the expected complete log likelihood reveals exactly one term for each IDR (the double sum $\sum_{i=1}^{n} \sum_{j=1}^{r}$ is over all n proteins over all $r$ IDRs in each protein) weighted by $\langle X_{ij} \rangle$, the expected value of the hidden variable. Thus, our model corresponds to an iteratively reweighted logistic regression problem at the IDR level, where the weights are the current guess of the hidden variable $\langle X_{ij} \rangle$, that represents which IDR is responsible for the function. We can therefore maximize the LASSO penalized version of this expected complete log likelihood, by adding an extra set of weights to the IRLS algorithm described above, so that the weights are:

$$w_{ij} = \langle X_{ij} \rangle \frac{1}{1 + e^{-Z_{ij} b}} \left( 1 - \frac{1}{1 + e^{-Z_{ij} b}} \right)$$

In each M-step, we do five iterations of this IRLS (on standard LASSO regression) to obtain new parameter values. We initialize the weights by setting them equal to one divided by the number of IDRs in the protein, so that they are downweighted equally.

In the E-step of the E–M algorithm, the expectations of the hidden indicator variables $\langle X_{ij} \rangle$ are calculated using Bayes' theorem as follows:

$$\begin{aligned} \langle X_{ij} \rangle &= P\left( X_{ij} = 1 | Y_i, Z_{ij}, b \right) = \frac{P\left( X_{ij} = 1 | Z_{ij}, b \right) P\left( Y_i | X_{ij} = 1, Z_{ij}, b \right)}{P\left( Y_i | Z_{ij}, b \right)} \\ &= \frac{Y_i P\left( Y_i = 1 | X_{ij} = 1, Z_{ij}, b \right) + (1 - Y_i) \left( 1 - P\left( Y_i = 1 | X_{ij} = 1, Z_{ij}, b \right) \right)}{\sum_{k} Y_i P\left( Y_i = 1 | X_{ik} = 1, Z_{ij}, b \right) + (1 - Y_i)(1 - P(Y_i = 1 | X_{ik} = 1, Z_{ik}, b))} \end{aligned}$$

Note that we have assumed no prior knowledge about which IDR in a protein is most likely to be responsible for the function (uninformative priors), that is $P\left( X_{ij} = 1 | Z_{ij}, b \right)$ is constant over $j$.

