## [Decision Letter]

**Acceptance summary:**

This research advance builds on the author's previous work on predicting the molecular function of intrinsically disordered from amino acid sequence and evolutionary dynamics. This is an emerging and important field and the author's contribution helps lay key foundations for how this question can be addressed. In addition novel scientific insight, a new online tool, the authors provide a detailed walk-through of the underlying statistical analysis performed.

**Decision letter after peer review:**

Thank you for submitting your article "Identifying molecular features that are associated with biological function of intrinsically disordered protein regions" for consideration by *eLife*. Your article has been reviewed by two peer reviewers, and the evaluation has been overseen by Michael Eisen as the Senior and Reviewing Editor. The reviewers have opted to remain anonymous.

The reviewers have discussed the reviews with one another and the Reviewing Editor has drafted this decision to help you prepare a revised submission.

The work presented offers a promising advancement for understanding protein function in general and of IDRs in particular. The manuscript presents the theoretical model development, experimental validation, and a protein engineering effort to further understand the results. A refreshing, pertinent discussion of potential shortcomings of the model is included throughout the manuscript. The manuscript is well-written but given its interdisciplinary nature, it appears overwhelmingly dense in details at times.

In general, it would be nice to have a less mathematical overview of the model workflow as a figure. What are the used features and the resulting predictions? Give more specific examples from the 82 features in Zarin et al., 2019.

Some specific comments raised by the reviewers:

Results/equations

"An interpretable regularized probabilistic model to predict function from IDR features" – I believe the LHS of Equation 1, P(Y=1|Z,b), does not match the description in the preceding paragraph. I think the issue is due to the lack of an “I” subscript for the “Y” variable. Since “I” corresponds to the specific protein, excluding the subscript and writing "P(Y=1)" seems to represent the probability that all proteins in the dataset have a particular functional annotation, while "P(Yi=1)" represents the probability that a particular protein has a particular functional annotation. This discrepancy causes the equivalency in Equation 1 to not be true ("all proteins have annotation" and "no proteins have annotation" are not the only two cases). This same issue extends into Equation 2. I don't believe there is any issue with the underlying approach of FAIDR, but it seems the equations (as written here) are not quite correct. It would be extremely useful to include a more detailed derivation in an Appendix in order to better explain the math behind the approach. Specifically if, this could be mathematically derived and explained in “laymans” terms this would, I think, significantly improve the utility of the manuscript, as for those less familiar with statistical inference/Bayesian analysis the mathematics as they stand basically just have to be trusted (which, having gone through they should, and clearly Dr. Moses is perhaps literally the world authority on this topic, BUT, never-the-less from a pedagogical standpoint it would do a great service to the field to provide a more accessible explanation of how the model is constructed).

Potential major concerns:

"IDR function can be predicted from protein-level annotations and IDR sequence properties" – As far as I can see the authors never indicate which specifics are used to derive their dataset, only that it is the "yeast proteomes". Depending on the source of the proteins in their dataset, it is possible that they have sequences with high similarity to one another in their dataset? This could be problematic. Since if they have two similar sequences with similar functional annotations, and the sequences are divided (since it's random) into the training and test sets, their predictions (and ROC curves) could be inflated and not a true measure of how accurate their method is. I.e., if the proteomic dataset they are using has already accounted for sequence similarity, then this is not a problem.

Results – The test of mitochondrial targeting is consistent with the authors analysis, but it's not clear to me if it's a particularly convincing demonstration of a novel approach, and perhaps the stated conclusions are a little strong given the evidence.

Some general concerns regarding the experiments:

There is no true negative control (i.e. completely redesign the sequence while maintaining pI and hydrophobicity), only the WT Cox15 pos control and various mutant sequences that are relatively similar (other than pI/hydrophobicity).

It is difficult to definitively conclude that the "sim IDR high pI" protein is localizing to the mitochondria from the image without concurrent labelling of mitochondria, only that it is localizing somewhere.

Not enough mutant sequences were tested to reach the conclusion that the localization must be due to high pI and/or high hydrophobicity of the sequence. For both primary mutants, less than 10% of the residues were mutated, so it's impossible to pinpoint what the exact cause of the loss of function is from. For example, given mitochondrial targeting signals have historically been described as amphipathic helices, is loss/gain of helicity a confounding variable? What about loss/gain of a cleavage site (as is commonly associated with targeting sequences).

All this said, the experiments do provide strong evidence for their conclusions, but I don't believe as high of confidence is warranted.

The features identified and then modified are known to be associated with mitochondrial targeting (in fact, represent how these sequences were historically identified, although predictions have now progressed somewhat).

To take a region that has a set of features that had already been identified as correlating with function and then modifying those features is – at least in principle – something that could have been done with the expected outcome without any of the authors methodologies. This of course does not invalidate the authors approach (and does strengthen the case) but it is in my mind not a well-defined “test”. This is not actually a huge problem in terms of publication, but perhaps something worth discussing. In defense of this section, the actually systematic approach the authors take to IDR mutation/evolution/rescue is great and I think a really useful blueprint for how one might test the importance of sequence features.

A sort of important question – is the Cox15 N-terminal IDR actually an IDR? It is not strongly predicted to be disordered (see http://d2p2.pro/view/sequence/up/P40086), and traditionally mitochondrial targeting sequences have been shown to form amphipathic helices (with a positive face). There may be experimental data demonstrating the disordered nature of this region, but it seems odd to select a demonstration example which it is (at least to me?) unclear if it meets the specific requirement for the approach to work. That said, a way to flip this is that as a prediction tool it is fundamentally assessing solvent-accessible residues, and any structural constraints imposed on those residues is perhaps less important than the mean-field chemical composition that the sequence provides. This might be a useful line of explanation (and has the added bonus of broadening the scope of the tool).

---

## [Author Response]

The work presented offers a promising advancement for understanding protein function in general and of IDRs in particular. The manuscript presents the theoretical model development, experimental validation, and a protein engineering effort to further understand the results. A refreshing, pertinent discussion of potential shortcomings of the model is included throughout the manuscript. The manuscript is well-written but given its interdisciplinary nature, it appears overwhelmingly dense in details at times.

We thank the reviewers for the supportive comments. We agree that the interdisciplinary nature of the paper requires us to be quite specific, and perhaps overwhelmingly so at times. We have tried to simplify the language throughout the manuscript, where possible, e.g. in the Abstract and Introduction. We have also reorganized the manuscript so that equations only appear in the Materials and methods.

In general, it would be nice to have a less mathematical overview of the model workflow as a figure. What are the used features and the resulting predictions? Give more specific examples from the 82 features in Zarin et al., 2019.

We have followed this suggestion. We have now included a less mathematical overview of the model workflow as Figure 1, including more specific examples from 82 features in Zarin et al., 2019, as suggested (e.g. mean CKII consensus sites, mean net charge, etc.).

Some specific comments raised by the reviewers:Results/equations"An interpretable regularized probabilistic model to predict function from IDR features" – I believe the LHS of Equation 1, P(Y=1|Z,b), does not match the description in the preceding paragraph. I think the issue is due to the lack of an “I” subscript for the “Y” variable. Since “I” corresponds to the specific protein, excluding the subscript and writing "P(Y=1)" seems to represent the probability that all proteins in the dataset have a particular functional annotation, while "P(Yi=1)" represents the probability that a particular protein has a particular functional annotation. This discrepancy causes the equivalency in Equation 1 to not be true ("all proteins have annotation" and "no proteins have annotation" are not the only two cases). This same issue extends into Equation 2. I don't believe there is any issue with the underlying approach of FAIDR, but it seems the equations (as written here) are not quite correct. It would be extremely useful to include a more detailed derivation in an Appendix in order to better explain the math behind the approach. Specifically if, this could be mathematically derived and explained in “laymans” terms this would, I think, significantly improve the utility of the manuscript, as for those less familiar with statistical inference/Bayesian analysis the mathematics as they stand basically just have to be trusted (which, having gone through they should, and clearly Dr. Moses is perhaps literally the world authority on this topic, BUT, never-the-less from a pedagogical standpoint it would do a great service to the field to provide a more accessible explanation of how the model is constructed).

The reviewer is right: we have now added the “I” subscripts where suggested.

As suggested, we have also added an Appendix with a more simple, laypersons derivation of the model. We now also describe the model more simply in the Results (under “A model that predicts protein function from molecular features in IDRs”), and have moved Equations 1 and 2 to the Materials and methods (under “The FAIDR statistical model”) so that no equations appear in the Results sections.

Potential major concerns:"IDR function can be predicted from protein-level annotations and IDR sequence properties" – As far as I can see the authors never indicate which specifics are used to derive their dataset, only that it is the "yeast proteomes". Depending on the source of the proteins in their dataset, it is possible that they have sequences with high similarity to one another in their dataset? This could be problematic. Since if they have two similar sequences with similar functional annotations, and the sequences are divided (since it's random) into the training and test sets, their predictions (and ROC curves) could be inflated and not a true measure of how accurate their method is. I.e., if the proteomic dataset they are using has already accounted for sequence similarity, then this is not a problem.

We thank the reviewer for raising this important point. Although the IDRs are highly diverged in general, there are a small fraction that show detectible sequence similarity. In order to quantify the sequence similarity in our training and test sets, we performed an all-by-all BLAST of the IDRs in each respective set. As we now describe in the Materials and methods (under “Evaluating FAIDR at the IDR level”), we find that 5% of the IDRs in our test set had significant sequence similarity (BLAST e-value less than 10^-5^) to IDRs in our training set. These were mostly comprised of paralogous proteins that were randomly split in our dataset. Further we report that sequence similarity explains up to 3% of our predictive power for mitochondrial targeting signals, and up to 10% of our predictive power for CDK targets. Hence, sequence similarity can account for at most only a small fraction of our classification results.

Results – The test of mitochondrial targeting is consistent with the authors analysis, but it's not clear to me if it's a particularly convincing demonstration of a novel approach, and perhaps the stated conclusions are a little strong given the evidence.

We take the reviewer’s point here. While we haven’t demonstrated any new biology in this analysis, we chose this example to illustrate the use of our new approach in guiding experimental design. We clarified the choice of a mitochondrial targeting signal as an example in this section.

Some general concerns regarding the experiments:There is no true negative control (i.e. completely redesign the sequence while maintaining pI and hydrophobicity), only the WT Cox15 pos control and various mutant sequences that are relatively similar (other than pI/hydrophobicity).

We agree with the reviewer here, and hence we never claim that hydrophobicity and pI are the only important features for mitochondrial localization. We have emphasized this the Discussion of the revised manuscript. Completely re-designing the sequence while maintaining pI and hydrophobicity would not be an applicable negative control here. We have tried to make this more clear in the Results and Discussion sections of the revised manuscript.

It is difficult to definitively conclude that the "sim IDR high pI" protein is localizing to the mitochondria from the image without concurrent labelling of mitochondria, only that it is localizing somewhere.

In order to more confidently conclude that the “sim IDR high pI” protein is localizing to the mitochondria, we repeated the localization experiments in the presence of DAPI staining, which allows us to visualize the nuclear and mitochondrial DNA. The colocalization of our GFP signal with the mitochondrial DNA staining is clearly visible in these experiments. We have included the results of this experiment in the relevant Results section (“Molecular features in the Cox15 IDR predictably affect mitochondrial targeting of Cox15”) and include the micrographs in Figure 4—figure supplement 1.

Not enough mutant sequences were tested to reach the conclusion that the localization must be due to high pI and/or high hydrophobicity of the sequence. For both primary mutants, less than 10% of the residues were mutated, so it's impossible to pinpoint what the exact cause of the loss of function is from. For example, given mitochondrial targeting signals have historically been described as amphipathic helices, is loss/gain of helicity a confounding variable? What about loss/gain of a cleavage site (as is commonly associated with targeting sequences).All this said, the experiments do provide strong evidence for their conclusions, but I don't believe as high of confidence is warranted.

Although we believe the definitive experiments suggested by the reviewer are beyond the scope of our study, we now provide some bioinformatics results to further support for our interpretation of the results. To test if the helicity or loss/gain of cleavage site could explain our results, we input our wildtype and mutant sequences into Mitofates, which predicts the helicity and cleavage sites for any given sequence. At least as far as the bioinformatics can be believed, neither the predicted helicity, nor the cleavage site, nor the recognition site can explain the phenotypic differences that we see when we make targeted mutations based on our model. We now include these results in Figure 4B (formerly Figure 3B). Although this is still indirect, we believe our interpretation of the data is consistent with all the evidence we have.

The features identified and then modified are known to be associated with mitochondrial targeting (in fact, represent how these sequences were historically identified, although predictions have now progressed somewhat).To take a region that has a set of features that had already been identified as correlating with function and then modifying those features is – at least in principle – something that could have been done with the expected outcome without any of the authors methodologies. This of course does not invalidate the authors approach (and does strengthen the case) but it is in my mind not a well-defined “test”. This is not actually a huge problem in terms of publication, but perhaps something worth discussing. In defense of this section, the actually systematic approach the authors take to IDR mutation/evolution/rescue is great and I think a really useful blueprint for how one might test the importance of sequence features.

We thank the reviewer for these comments. The reviewer is right in that we strategically chose an easily observable phenotype which has been heavily studied, and in which we could easily manipulate sequence features that have (in some cases) been used to characterize these regions. As suggested by the reviewer, we now provide more context for why we chose this as a test case in the first paragraph of the relevant Results section (“Molecular features in the Cox15 IDR predictably affect mitochondrial targeting of Cox15”) and in the Discussion.

A sort of important question – is the Cox15 N-terminal IDR actually an IDR? It is not strongly predicted to be disordered (see http://d2p2.pro/view/sequence/up/P40086), and traditionally mitochondrial targeting sequences have been shown to form amphipathic helices (with a positive face). There may be experimental data demonstrating the disordered nature of this region, but it seems odd to select a demonstration example which it is (at least to me?) unclear if it meets the specific requirement for the approach to work. That said, a way to flip this is that as a prediction tool it is fundamentally assessing solvent-accessible residues, and any structural constraints imposed on those residues is perhaps less important than the mean-field chemical composition that the sequence provides. This might be a useful line of explanation (and has the added bonus of broadening the scope of the tool).

We have now provided more context on mitochondrial presequences. We explain in the Discussion section that although mitochondrial targeting sequences form helices upon binding to their target receptors, they sample dynamic conformations (like many disordered regions). Following the reviewer’s suggestion, we note that our approach can be applied to any protein region, but is particularly useful for those protein regions which are not amenable to standard methods of sequence analysis.